# Validity of heart rate measurements in wrist-based monitors across skin tones during exercise

**Stanley Hughwa Hung**[1,2], **Kelsey Serwa**[1], **Gillian Rosenthal**[1], **Janice J. Eng**[1,2]*

**1** Department of Physical Therapy, The University of British Columbia, Vancouver, British Columbia, Canada, **2** Rehabilitation Research Program, Centre for Aging SMART, Vancouver Coastal Health Research Institute, Vancouver, British Columbia, Canada

* janice.eng@ubc.ca

**Data Availability Statement:** The study data and statistical analysis code used for data analysis are publicly available in the Borealis Dataverse Repository at the following DOI: https://doi.org/10.5683/SP3/LLD99A.

## Abstract

### Purpose

To evaluate the accuracy of a wrist-based heart rate (HR) monitor at different exercise intensities across different skin tones.

### Methods

Using a cross-sectional design, we compared HR measures from the wrist-based photo-plethysmography Fitbit Charge 5 to the Polar H10 chest strap at rest and during the YMCA Protocol using a recumbent cycle ergometer. Participant were grouped into three skin tone categories: light (Fitzpatrick Scale Skin Types 1+2), medium (Types 3+4), and darker skin tone (Types 5+6). HR measures using the Polar chest strap during the exercise test were categorized as <40%, 40–60%, or >60% HR reserve (HRR). Absolute error in beats per minute (bpm) between the two devices was calculated for each measure. A linear mixed effects model was used to assess interaction effects between skin tone and exercise intensity, with participants as the random effect. Bland-Altman plots were used for visual analyses.

### Results

Twenty-five participants [mean (SD): 25.8 (1.9) years old; 64% female] were included with 495 observations of simultaneous Fitbit and Polar HR recordings collected during exercise. During exercise, we observed a statistically significant interaction effect between skin tone and exercise intensity. Compared with light skin tone at <40% HRR, mean error was greater for medium skin tone at >60% HRR [mean error (95%CI): 11.8 (5.6–17.9) bpm, p<0.001] and darker skin tone at 40–60% HRR [7.6 (1.7–13.5) bpm, p = 0.011] and >60% HRR [11.7 (5.3–18.0) bpm, p<0.001].

### Conclusion

HR measurement error using a wrist-based device was greater with increasing exercise intensity for people with darker skin tones.

**Funding:** This study was supported by the Canada Research Chairs Program (to J. Eng), Michael Smith Health BC Postdoctoral Award (to S. Hung), and the Canadian Institutes of Health Research Foundation Grant (FDN 143340; to J. Eng). The funders had no role in study design, data collection and analysis, decision to publish, or preparation of the manuscript.

**Competing interests:** The authors have declared that no competing interests exist.

## Introduction

The benefits of aerobic exercise and physical activity for chronic disease prevention and improving quality of life after a chronic disease diagnosis are well-established [1, 2]. These benefits are often associated with exercise and physical activity at least at moderate intensity [3], which is typically measured as a percentage of heart rate (HR) reserve (HRR) [4]. The popularity of wearable fitness trackers to monitor HR has increased in recent years for consumers in the general public, healthcare, and research settings [5], including monitoring HR for exercise prescription or estimating oxygen consumption during exercise testing [6]. Electrocardiogram (ECG), while considered the gold standard for measuring HR, is expensive and difficult for field-use given the need for multiple wires and electrodes placed across the thorax [7, 8]. For these reasons, wireless, ECG-based chest strap HR monitors have grown in popularity and are valid (i.e., accuracy of measuring heart rate) alternatives to ECG [7]. These commercially available heart rate chest straps have strong validity against ECG in healthy adults [9–11]. Despite its validity, chest strap HR monitors have some disadvantages. They can be uncomfortable when worn for long durations (example, chest pressure and friction), difficult to don for some populations (example, people with arm impairments), and require regular, thorough cleaning within healthcare settings [12]. Wrist-based HR monitors may offer a more convenient and comfortable option for measuring HR [12].

Wrist-based HR monitors use photoplethysmography (PPG) technology, an optical technique that uses light absorption beneath the skin to non-invasively measure changes in blood flow volume between systolic and diastolic phases of the cardiac cycle [13, 14]. While more convenient and comfortable to wear compared to ECG, the validity of these devices has been questioned. Accurate measurements of HR using PPG devices is highly dependent on PPG signal accuracy, which can be influenced by multiple factors, including type of sensors used (example, shape of sensors and light intensity), post-processing algorithms used, the deformation of the skin when device is worn, vascular aging (example, arterial stiffness), and excessive movement [15]. A systematic review (32 studies, 1085 participants, 16 different devices) concluded that only certain wrist-based HR monitors have been tested with acceptable validity, and the accuracy can vary widely depending on the model (up to 20% mean percentage error) [16]. One important source of inaccuracy includes the potential impact of different skin tones on the signals derived from PPG technology [14]. Melanin, a substance responsible for skin pigmentation, tends to absorb green light and reduces light reflection, which may affect the signal accuracy [17]. Another systematic review (10 studies, 469 participants, 26 different devices) rigorously examined the available literature on the impact of skin tone on the accuracy of HR measurements using wrist-based PPG devices [18]. The authors concluded that there are conflicting results in HR measurement accuracy across different skin tones, with four studies observing lower accuracy with darker skin tones, four observing no difference, and two having mixed results [18]. The authors attributed the mixed results partly due to inadequate sample sizes, use of multiple devices for comparison, and inconsistent methods with categorizing skin tones (example, non-standardized self-reported race and ethnicity) [18]. One potential confounding factor in these studies is the level of exercise intensity. Higher exercise intensities have been shown to reduce accuracy of HR measurements, due to factors such as arm motion artifact, sweat accumulation under the PPG sensors, variations in contact pressure between the skin and device sensors, and peripheral vascular resistance changes with increased exercise intensity [19–23]. The combined effects of skin tone and increased exercise intensity may further decrease HR accuracy. Studies have examined the difference in HR accuracy between rest and exercise conditions or with increased intensities alone, without skin tone comparisons [17, 24–29]. One study descriptively reported no skin tone differences in HR

accuracy with increasing exercise intensity, without reporting statistical analyses [30]. There-fore, the effect of skin tone on HR measurement accuracy using PPG technology remains inconclusive and the interaction between skin tone and progressive increase in exercise inten-sity has not been examined. This potential source of inaccuracy could have important implica-tions for use across ethnically-diverse populations [18, 24]. In clinical settings, such as people with cardiovascular disease, monitoring HR during exercise is recommended to ensure people are safely exercising at HR below the onset of ischemic symptoms and at an adequate intensity to maximize exercise adaptations and benefits [31]. People with darker skin tones (example, Black and Hispanic ethnic backgrounds) are at higher risk of cardiovascular diseases [32], and inaccurate HR measurements using PPG-based devices would result in potential health dispar-ities with exercise safety and treatment efficacy in this clinical population. Given these poten-tial accuracy issues, international wearable HR device experts recommend continued validation testing on that the effect of skin tone on HR measurement accuracy.

The primary aim of this study was to evaluate the accuracy of wrist-based HR monitors at different exercise intensities across different skin tones. We hypothesized that accuracy of measurements would be lower with darker skin tones, with progressively lower accuracy at higher exercise intensities, compared to lighter skin tones. The secondary aim of this study was to evaluate the accuracy of the wrist-based HR monitors at rest across different skin tones. We hypothesize that accuracy of measurements will not differ between different skin tones.

## Materials and methods

This study used a cross-sectional design. Ethical approval was obtained from the University of British Columbia Behavioural Research Ethics Board (H22-02543). All participants provided written informed consent.

### Participant recruitment

Participants were recruited between February and May 2023 with purposive, convenience sampling to ensure people across all skin tone categories were included. Broadcast emails to local community organizations and word-of-mouth were used for recruitment. People were eligible if they were adults aged 18–30 years with no history of cardiovascular disease and cleared for exercise using The Physical Activity Readiness Questionnaire (PAR-Q). People who reported having any known medical conditions (example, arrythmias) or medications (example, beta-blockers) that affected cardiovascular function were excluded.

### Devices

The Fitbit Charge 5 (FB5; Fitbit, Inc., California, USA) was selected as the wrist-based HR monitor and was compared to the Polar H10 (POL; Polar Electro Oy, Kempele, Finland). The POL is a commercially available HR chest strap with strong agreement and small bias against ECG during rest and exercise (bias -0.1 to -0.2 beats per minute [bpm], limits of agreement -0.7 to 0.5 bpm, $R^2$ = 1.00, standard estimated error = 0.26, p<0.001) [9]. Given its strong validity against ECG, international experts recommend the use of HR chest straps as an appro-priate criterion measure for HR [11]. Furthermore, HR chest straps use electrical signals simi-lar to ECG to estimate HR, and is unlikely to be influenced by skin tone [11, 18]. The FB5 was selected as wrist-based HR monitor because it is relatively inexpensive (~180 CAD) and com-mercially available, making it a likely option to be used by the general public and within clini-cal settings. The FB5 that uses green-light PPG technology and has been previously tested with 6.2% mean absolute error with ECG [33]. The FB5 was fastened and fitted according to manu-facturer's instructions on the left wrist [34]. Handedness did not influence device fitting as

previous literature found no significant difference in accuracy between the dominant versus non-dominant arm [35].

## Study procedures

**Skin tone categorization.** After eligibility screening, potential participants completed a skin tone screen via Zoom Teleconferencing Software (Zoom Video Communications, Inc., California, USA) using the Fitzpatrick Scale, which categorizes skin tones across 6 types based on colour and ultraviolet radiation response [36]. Table 1 describes the 6 Fitzpatrick Scale skin tone categories [37]. The Fitzpatrick Scale has been validated against melanin concentration [38], and is commonly used in research investigating the impact of skin tone on HR measurement accuracy [18]. Purposive sampling ensured a minimum of 4 participants in each Fitzpatrick Skin Type. Participants categorized with the desired skin tone were invited for an in-person study visit. To ensure accurate skin tone categorization, a second researcher independently rated participants at the in-person visit. A third researcher's rating resolved disagreements, if any.

**Resting and exercise testing heart rate.** A single resting HR measurement was collected from both devices by taking the HR observed at the end of five minutes of quiet sitting. Participants then completed an exercise session using the SCIFIT-ISO1011 Recumbent Bike (Life Fitness, Illinois, USA). The bike seat was adjusted to allow for 30° knee flexion at the most extended position. The exercise session included a 5-minute warmup [25 watts (W)], followed by an exercise test, and a 5-minute cooldown (25W). The YMCA Cycle Ergometer Protocol [39] was used for exercise testing. Participants started at 50W maintained at 50 revolutions per minute, with incremental power progressions that depended on participants' heart rate at the end of each stage (example, at the end of the first stage at 50W, stage 2 would be 150W for HR <90 bpm, 125W for HR 90–105 bpm, and 100W for HR >105 bpm). Each stage lasted 3 minutes. If the test was not terminated after 3 stages, each subsequent stage involved an increase of 25W until test termination. The exercise test was terminated when the participant reached 80% HRR or by voluntary termination (example, lower limb fatigue). The recumbent bike was selected to minimize motion artefact from arm movement [11, 22, 40]. Participants were instructed to relax their hands at their sides throughout testing, as gripping handles and upper body movement may contribute to measurement error [11]. HR was recorded from each device every minute during the warmup, exercise test, and cooldown. To ensure HR readings from the POL and FB5 devices were recorded at the same time, one investigator was assigned to record HR data from each device and a central timer was used to ensure readings were recorded at the same time. A verbal countdown facilitated the two devices being started at the same time (within a second). Maximum HR was estimated [HRmax = 206.9 - (0.67 x age)]

**Table 1. The Fitzpatrick Scale description for skin tone categorization\*.**

| Fitzpatrick Skin Types | Description |
| --- | --- |
| 1 | Always burns, never tans, and sensitive to ultraviolet light exposure. |
| 2 | Burns easily, and tans minimally. |
| 3 | Burns moderately, and gradually tans to light brown |
| 4 | Burns minimally, and always tans to moderate brown |
| 5 | Rarely burns, and tans to dark |
| 6 | Never burns, high pigmentation, and least sensitive to changes |

\*Adapted from the Surgeon General's Call to Action to Prevent Skin Cancer

[41], and used with resting HR to calculate HR reserve (HRR, Maximum HR minus resting HR [42]. Each study visit lasted approximately 45 minutes.

## Statistical analysis

Statistical analyses were conducted using the R computing environment (version 4.2.2., R Core Team, 2022), with 0.05 alpha (level of significance; 5% probability of incorrectly rejecting the null hypothesis or falsely detecting an effect when there is no true effect [Type 1 Error]). Participants were grouped into three skin tone categories: light skin tone (Fitzpatrick Skin Types 1 and 2), medium skin tone (Types 3 and 4), and dark skin tone (Types 5 and 6). Each HR measure from each device was treated as a repeated measure for each participant. Each HR measure with the POL during exercise (i.e., warmup, the YMCA test, and cooldown) was categorized as <40% (low intensity exercise), 40–60% (moderate intensity), or >60% HRR (high intensity). The error in HR measures was calculated by taking the absolute difference between the POL and FB5. Participant characteristics and outcomes were summarized in means, standard deviations, and frequencies, where applicable.

Our primary analysis examined the accuracy of the FB5 during exercise. We used a linear mixed-effects model (*lme4* package [43]); HR error was the dependent variable, and fixed effects were exercise intensity (low, moderate and high) and skin tone (light, medium, and dark), and the interaction effects between exercise intensity and skin tone. We included participants as a random effect. Using the GLIMMPSE Software (version 3.1.2, Colorado, USA) [44] for linear mixed-effects models, we determined that a minimum sample size of 24 participants (4 per Fitzpatrick Skin Type) was required to detect a clinically meaningful difference of 10 bpm (i.e., for monitoring exercise progression and avoiding potential adverse events) [31] with a standard deviation of 6.8 bpm [17], repeated-measure correlation of 0.5, and 80% power (probability of correctly rejecting the null hypothesis; 80% chance of detecting statistical significance when a true effect is present) at 0.05 alpha. To account for 10% attrition (3 participants), the target sample size was increased to 27 participants. We tested the linear mixed effects model assumptions (see S1 File for assumptions results). We observed mild heteroskedasticity of variance (S1 Fig in S1 File) and mild positive skewness in our fixed effects residual distribution (S2 and S3 Figs in S1 File). Linear mixed effects models are robust to violation to distributional assumptions and heteroscedasticity and remains an appropriate model for our study data [45–47].

Bland-Altman plots were used to examine the mean differences and 95% limits of agreement for HR measures between POL and FB5 during exercise for the three skin tone categories.

For our secondary analyses, we examined the accuracy in resting HR measurements across different skin tones using a linear regression model (*lme4* package [43]) where HR error was the dependent variable and skin tone (light, medium, and dark) was the independent variable.

## Results

### Participants

Forty-one individuals expressed interest, 37 were screened, and 27 were recruited (see Fig 1 for the recruitment process flowchart). Data from 2 participants were removed from the analysis. One participant (Fitzpatrick Skin Type 3, female) reported improper device fitting after study completion, and the other participant (Fitzpatrick Skin Type 3, male) experienced a recumbent bike mechanical failure and did not return for re-testing. Twenty-five participants (females = 16) were included in the analyses (see Table 2 for characteristics), with 495 observations of simultaneous FB5 and POL HR recordings collected during exercise. The exercise

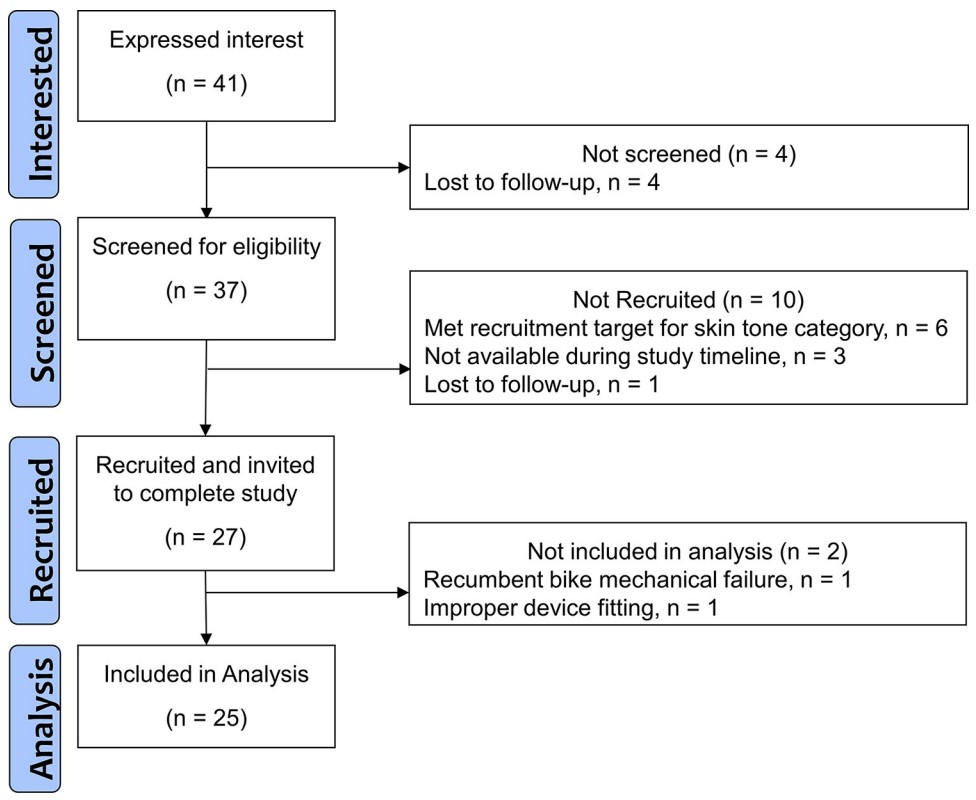

**Fig 1. Study recruitment process flowchart.**

testing duration was a mean 9.6 minutes (SD: 2.1), acquiring a mean 39.6 (SD: 4.4) observations per participant. Two participants did not achieve the 80% HRR target during exercise testing due to muscle fatigue. Eight participants were in the light skin tone group, 8 for medium skin tone, and 9 for dark skin tone. Four participants from each Fitzpatrick Skin Type were included in the analysis, except for Skin Type 5 with 5 participants. One HR reading with the FB5 for one participant (Fitzpatrick Skin Type 1, female) was missing due to device syncing error, and this observation's comparison POL value was excluded from the analyses.

## Heart rate error by skin tone groups

Tables 3 and 4 summarize the mean error of HR measures at rest and during different exercise intensities, respectively. For resting measures, the mean error between the POL and FB5 for all

**Table 2. Participant characteristics.**

| Characteristics | All Participants (n = 25) | Light Skin Tone (n = 8) | Medium Skin Tone (n = 8) | Dark Skin Tone (n = 9) |
|---|---|---|---|---|
| Age, years, mean ± SD | 25.8 ± 1.9 | 26.4 ± 2.3 | 25.3 ± 2.3 | 25.9 ± 1.3 |
| Female, n (%) | 16 (64%) | 8 (100%) | 5 (62.5%) | 3 (37.5%) |
| Height, cm, mean ± SD | 172.3 ± 7.4 | 172.3 ± 4.2 | 170.7 ± 8.4 | 173.6 ± 9.3 |
| Weight, kg, mean ± SD | 70.6 ± 13.6 | 66.6 ± 5.6 | 68.1 ± 13.4 | 77.6 ± 18.0 |
| Body mass index, kg/m$^2$, mean ± SD | 23.7 ± 3.4 | 22.4 ± 1.2 | 23.2 ± 2.8 | 25.5 ± 4.9 |

SD = standard deviation; cm = centimeters; kg = kilograms; m = metres; light skin tone = Fitzpatrick Skin Types 1 and 2; medium skin tone = Fitzpatrick Skin Types 3 and 4; dark skin tone = Fitzpatrick Skin Types 5 and 6.

**Table 3. Absolute error of resting heart rate measurements organized by skin tone groups.**

| Skin Tone | Observations, n | Absolute Error of Heart Rate, mean ± SD, bpm |
|---|---|---|
| All | 25 | 2.8 ± 2.7 |
| Light | 8 | 2.3 ± 1.6 |
| Medium | 8 | 3.0 ± 3.3 |
| Dark | 9 | 3.0 ± 3.0 |

SD = standard deviation; light skin tone = Fitzpatrick Skin Types 1 and 2; medium skin tone = Fitzpatrick Skin Types 3 and 4; dark skin tone = Fitzpatrick Skin Types 5 and 6.

participants was 2.8 bpm (SD: 2.6) and were similar across all skin tone groups (Table 3). During exercise, the mean error remained relatively stable across all exercise intensities for light skin tone (approximately 4 bpm) (Table 4). However, for participants with medium and dark skin tones, the mean error was larger as exercise intensity increased, with the greatest mean error observed in exercise intensity >60% HRR with the dark skin tone group (mean 16.5 bpm). This was more than four times the error of the light skin tone group at >60% HRR (mean 3.5 bpm).

## Accuracy of resting and exercise heart rate measurements

We observed no difference in resting HR error across the three skin tone groups (Table 5). Table 6 summarizes the linear mixed-effects model results examining the effects of skin tone and exercise intensity on HR error. No independent effects of skin tone nor exercise intensity on error were observed. However, statistically significant interaction effects were observed between medium skin tone at >60% HRR, and dark skin tone at 40–60% HRR and >60% HRR. Specifically, compared with light skin tone at <40% HRR, an increase in error of 11.8 bpm (p<0.001) for >60% HRR was observed for medium skin tone, and an increase in error of 7.6 bpm (p = 0.011) and 11.6 bpm (p<0.001) for 40–60% HHR and >60% HHR for dark skin tone, respectively.

The Bland-Altman plots (Fig 2) illustrated larger mean differences, greater limits of agreement, and a greater number of observations outside of the limits of agreement with darker

**Table 4. Absolute error of heart rate measurements at different exercise intensities organized by skin tone groups.**

| Skin Tone | Exercise Intensity | Observations, n | Absolute Error of Heart Rate, bpm, mean ± SD |
|---|---|---|---|
| Light | All | 161 | 4.2 ± 6.1 |
| | <40% HRR | 94 | 4.3 ± 5.4 |
| | 40–60% HRR | 32 | 4.8 ± 8.5 |
| | >60% HRR | 35 | 3.5 ± 5.2 |
| Medium | All | 165 | 5.7 ± 11.8 |
| | <40% HRR | 99 | 3.0 ± 4.4 |
| | 40–60% HRR | 38 | 6.0 ± 9.0 |
| | >60% HRR | 28 | 14.9 ± 23.4 |
| Dark | All | 169 | 10.2 ± 16.2 |
| | <40% HRR | 101 | 6.7 ± 8.4 |
| | 40–60% HRR | 42 | 14.6 ± 19.5 |
| | >60% HRR | 26 | 16.5 ± 26.5 |

SD = standard deviation; HRR = heart rate reserve; light skin tone = Fitzpatrick Skin Types 1 and 2; medium skin tone = Fitzpatrick Skin Types 3 and 4; dark skin tone = Fitzpatrick Skin Types 5 and 6.

**Table 5. Linear regression model for the effect of skin tone on absolute error of heart rate for the Polar H10 and Fitbit Charge 5.**

| | Absolute Error of Heart Rate, bpm | | |
|---|---|---|---|
| *Predictors* | *Estimates* | *95% Confidence Intervals* | *p* |
| (Intercept) | 2.3 | 0.2, 4.3 | **0.030** |
| Medium skin tone | 0.8 | -2.1, 3.6 | 0.590 |
| Dark skin tone | 0.8 | -2.0, 3.5 | 0.579 |
| Observations | 25 | | |
| $R^2$ / $R^2$ adjusted | 0.018 / -0.071 | | |

Values were compared to light skin tone. Bold values indicate p < 0.05. Light skin tone = Fitzpatrick Skin Types 1 and 2; medium skin tone = Fitzpatrick Skin Types 3 and 4; dark skin tone = Fitzpatrick Skin Types 5 and 6; bpm = beats per minute. Note: the intercept indicates the absolute error in heart rate without including effects of other skin tones (i.e., the effect of light skin tone only).

skin tones during exercise, suggesting reduced accuracy with darker skin tones. The FB5 tended to underestimated HR. Furthermore, the magnitude of mean differences and number of observations outside of the limits of agreement increased with greater mean HR, suggesting lower accuracy with increased exercise intensity.

## Discussion

While no accuracy issues were observed at rest, we observed differences in measurement error with increasing exercise intensities with darker skin tones. One of the reasons why the

**Table 6. Linear mixed effect model results for the effects of skin tone and exercise intensity on absolute error of heart rate for the Polar H10 and Fitbit Charge 5.**

| | Absolute Error of Heart Rate, bpm | | |
|---|---|---|---|
| *Predictors* | *Estimates* | *95% Confidence Interval* | *p* |
| (Intercept) | 4.3 | 0.1, 8.43 | **0.044** |
| Medium skin tone | -1.2 | -7.0, 4.7 | 0.694 |
| Dark skin tone | 2.4 | -3.3, 8.2 | 0.409 |
| 40–60% HRR | 1.0 | -3.3, 5.4 | 0.641 |
| >60% HRR | -0.6 | -4.8, 3.6 | 0.784 |
| Medium skin tone X 40–60% HRR | 1.9 | -4.0, 7.8 | 0.537 |
| Dark skin tone X 40–60% HRR | 7.6 | 1.7, 13.5 | **0.011** |
| Medium skin tone X >60% HRR | 11.8 | 5.6, 17.9 | **<0.001** |
| Dark skin tone X >60% HRR | 11.7 | 5.3, 18.0 | **<0.001** |
| **Random Effects** | | | |
| $\sigma^2$ | 114.19 | | |
| $\tau_{00\ participants}$ | 26.10 | | |
| Intraclass Correlation Coefficient | 0.19 | | |
| $N_{participants}$ | 25 | | |
| Observations | 495 | | |
| Marginal $R^2$ / Conditional $R^2$ | 0.127 / 0.290 | | |

Bold values indicate p < 0.05. HRR = heart rate reserve; bpm = beats per minute; light skin tone = Fitzpatrick Skin Types 1 and 2; medium skin tone = Fitzpatrick Skin Types 3 and 4; dark skin tone = Fitzpatrick Skin Types 5 and 6. Values were compared to light skin tone and exercise intensity <40% HRR. Note: the intercept indicates the absolute error in heart rate without including effects of other skin tones (i.e., the effect of light skin tone only). For random effects: an $\sigma^2$ value of 114.19 bpm indicates a 10.69 bpm (square root of 114.19) of absolute error explained by within-participant variance; a $\tau_{00\ participants}$ value of 26.10 bpm indicates 5.11 bpm (square root of 26.10) between-participant variance; a 0.19 intraclass correlation coefficient indicates that 19% of the total variance of absolute error of heart rate is attributed to between-participant variance.

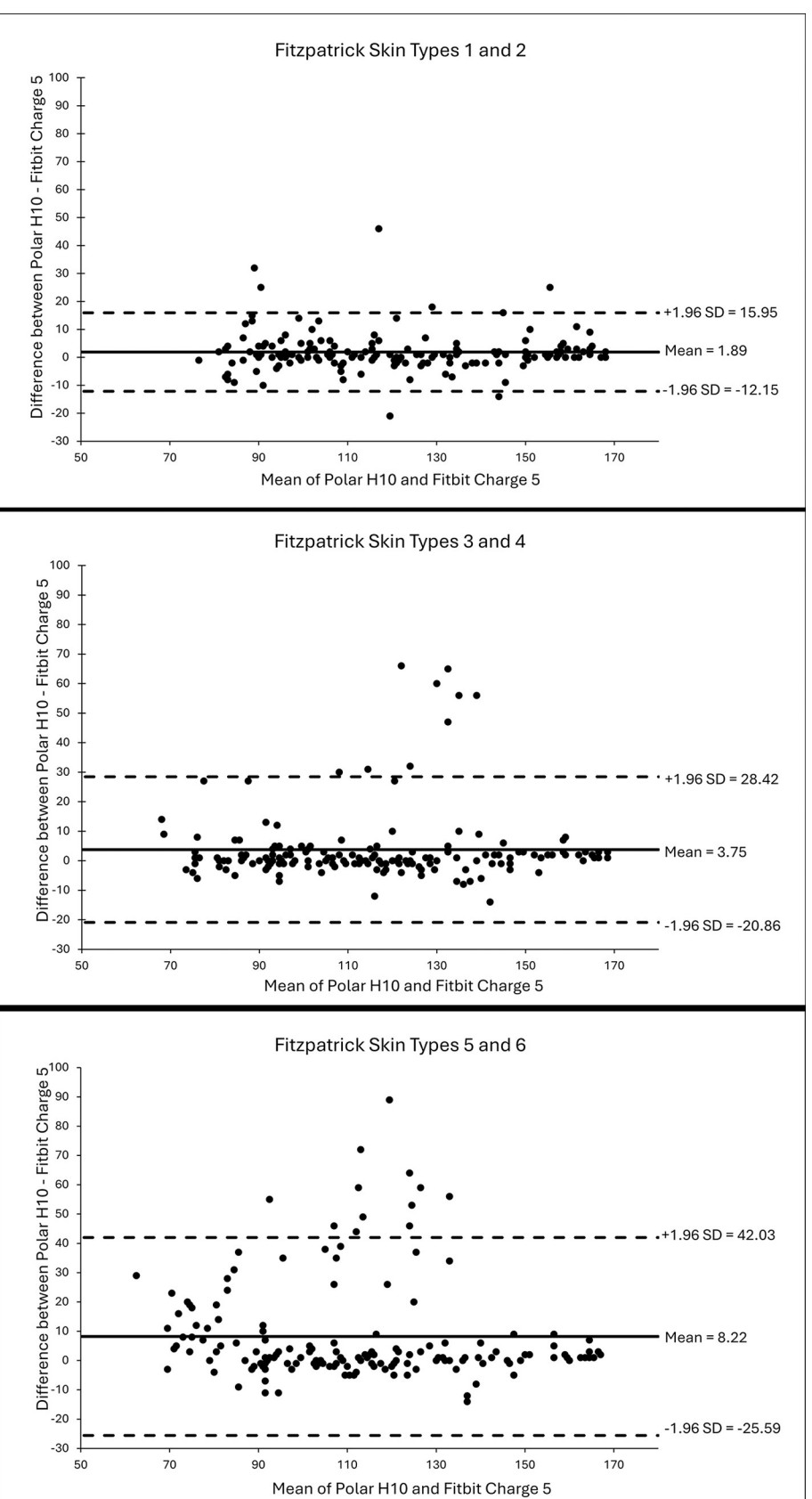

**Fig 2. Bland-Altman plots of heart rate measurement differences (bpm) during exercise testing between the Polar H10 and Fitbit Charge 5.** (A) Light Skin Tone (Fitzpatrick 1 and 2); (B) Medium Skin Tone (Fitzpatrick Types 3 and 4); and (C) Dark Skin Tone (Fitzpatrick 5 and 6). Solid lines represent mean differences. Dashed lines represent limits of agreement.

previous literature may have found conflicting results on HR accuracy and skin tone [18] is because prior studies did not control for levels of exercise intensity. We provide novel contributions by reporting interaction effects observed between skin tone and exercise intensity. Specifically, compared with light skin tones at <40% HRR, we observed increased measurement error (mean error 11.8 bpm) as exercise intensity increased beyond 60% HRR for participants with medium skin tones, but not for 40–60% HRR. Furthermore, significantly greater mean error was already observed for dark skin tones at 40–60% HRR (7.6 bpm), with even greater mean error beyond 60% HRR (11.7 bpm). This may indicate that people with darker skin tones are more susceptible to the effects of increased exercise intensity on accuracy.

Previous studies have examined HR measurement validity of Fitbit Charge devices and reported increased errors with higher HR during exercise, but did not account for skin tone [33, 48]. Motion artefact is considered a primary factor associated with HR measurement error in PPG devices during exercise [22]. We used a recumbent cycle to minimize upper body motion and its subsequent impact on motion artefact [11, 40]. Nonetheless, we continued to observe measurement errors with increasing exercise intensity, predominantly in people with darker skin tones. The reduction in accuracy with increasing exercise intensity may be explained through the mechanisms by which physiological changes during exercise are detected in PPG signals. HR measurements are derived by detecting the periodical changes in PPG signals corresponding to changes in blood flow during the systolic and diastolic phases of the cardiac cycle [13, 14]. As exercise intensity increases, there is a corresponding increase in blood pressure and reduction in vascular resistance to increase blood flow to meet oxygen demands [49]. The accuracy of HR measurements are dependent on accurate detection of these periodical physiological changes through PPG signals [50]. Greater melanin concentrations (i.e., darker skin tones) are associated with greater green light absorption [17], which may impact the optical sensors' ability to detect PPG signals changes with increased HR, potentially leading to measurement errors. Further research would be required to elucidate the mechanisms associated with the interaction effects between skin tone and exercise intensity on HR measurement accuracy.

Given the increasing popularity of wearable sensors for monitoring exercise progression and safety, their accuracy with HR measurements is crucial. This is especially important for clinical populations such as individuals with cardiovascular diseases, where people with darker skin tones (example, Black and Hispanic ethnic backgrounds) are at higher risk of cardiovascular disease [32] and exercise therapy is a primary option to reduce cardiovascular disease risk [24]. The measurement errors of 16 bpm during exercise observed in people with darker skin tones in the current study, compared with 4 bpm in lighter skin tones, is clinically significant. The American Heart Association recommends monitoring HR response during exercise, and staying approximately 10 bpm below the onset of ischemic symptoms (example, angina) for cardiac patients with an ischemic response [31]. Furthermore, international guidelines recommend exercising at least at moderate intensity ($\geq$40% HRR) for cardiovascular benefits from aerobic exercise [51, 52]. Therefore, an error of 16 bpm, mostly underestimations, may result in inadequate safety monitoring and exercise dosing, a reduction in treatment efficacy, and potentially an increased risk of adverse events [24]. Given these findings, developers should consider strategies to correct inaccuracies associated with skin tone. For example, correction algorithms have been developed to reduce the impact of motion artefact on HR

measurement accuracy during exercise [22, 53]. Similar correction algorithms may be a viable solution for skin tone. Furthermore, developers need to ensure comprehensive testing of their devices on a diverse population to enhance generalizability. Our results can be generalized to young adults without comorbidities; future studies should examine older and clinical populations to investigate the potential effects of aging and changes in cardiovascular function [54].

This study had several strengths. We used two independent raters to increase the reliability of our Fitzpatrick Scale categorization and included participants across all skin tone categories. We aimed to minimize the impact of motion artefact [22] by reducing arm movement or need for gripping handles using a recumbent bike, and the impact of contact pressure variations [21] by following the manufacturer watch-fitting guidelines. The study had several limitations. We did not use ECG, which is considered the gold standard for HR measurement [7, 8], which may have influenced our reference HR values and the study results. However, chest straps provide strong validity against ECG, are inexpensive, and easier to use during exercise compared with ECG [7, 8, 55]. The small study sample size may have increased the possibility of Type 2 Error for detecting potential significant differences in moderate intensity (40–60% HRR) between light and medium skin tone. Finally, this study only included one wrist-based HR monitor device in the analyses. Testing with multiple devices would increase generalizability across different design specifications known to impact PPG recordings, including different processing algorithms, sampling rate, and light source positions and intensity [13].

We observed significant HR measurement errors with increasing exercise intensity in those with darker skin tones, but not for lighter skin tones. Device developers should consider solutions to correct errors associated with skin tone differences. Larger studies with multiple devices in different populations would further elucidate the interaction between skin tone and exercise intensity on HR measurement using wrist-based devices.

## Supporting information

**S1 File. Statistical analysis assumptions testing for linear mixed effects model.** Results for the model assumptions testing for the linear mixed effects model.
(PDF)

## Acknowledgments

The study team would like to thank the study participants and Asahi Ng, Jackson Liu, and Nadine Akbarali for contributing to participant recruitment and data collection.

## Author Contributions

**Conceptualization:** Janice J. Eng.

**Data curation:** Stanley Hughwa Hung, Kelsey Serwa, Gillian Rosenthal.

**Formal analysis:** Stanley Hughwa Hung, Kelsey Serwa, Gillian Rosenthal.

**Funding acquisition:** Janice J. Eng.

**Investigation:** Stanley Hughwa Hung, Kelsey Serwa, Gillian Rosenthal.

**Methodology:** Stanley Hughwa Hung, Janice J. Eng.

**Project administration:** Stanley Hughwa Hung, Kelsey Serwa, Gillian Rosenthal, Janice J. Eng.

**Resources:** Janice J. Eng.

**Supervision:** Stanley Hughwa Hung, Janice J. Eng.

**Writing – original draft:** Stanley Hughwa Hung.

**Writing – review & editing:** Kelsey Serwa, Gillian Rosenthal, Janice J. Eng.

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
