## [Decision Letter · Decision Letter 0]

3 Sep 2024

PONE-D-24-32801Validity of Heart Rate Measurements in Wrist-Based Monitors Across Skin Tones During ExercisePLOS ONE

Dear Dr. Eng,

Thank you for submitting your manuscript to PLOS ONE. After careful consideration, we feel that it has merit but does not fully meet PLOS ONE’s publication criteria as it currently stands. Therefore, we invite you to submit a revised version of the manuscript that addresses the points raised during the review process.

We look forward to receiving your revised manuscript.

Kind regards,

Agnese Sbrollini

Academic Editor

PLOS ONE

“Funding for the current study was supported by the Canada Research Chairs Program (to J Eng), Michael Smith Health BC Postdoctoral Award (to S Hung), and the Canadian Institutes of Health Research Foundation Grant (FDN 143340; to J Eng).”

“Funding for the current study was supported by the Canada Research Chairs Program (to J Eng), Michael Smith Health BC Postdoctoral Award (to S Hung), and the Canadian Institutes of Health Research Foundation Grant (FDN 143340; to J Eng). The study team would like to thank the study participants and Asahi Ng, Jackson Liu, and Nadine Akbarali for contributing to participant recruitment and data collection.”

“Funding for the current study was supported by the Canada Research Chairs Program (to J Eng), Michael Smith Health BC Postdoctoral Award (to S Hung), and the Canadian Institutes of Health Research Foundation Grant (FDN 143340; to J Eng).”

Reviewers' comments:

Reviewer's Responses to Questions

**Comments to the Author**

1. Is the manuscript technically sound, and do the data support the conclusions?

Reviewer #1: Yes

Reviewer #2: Yes

2. Has the statistical analysis been performed appropriately and rigorously? 

Reviewer #1: Yes

Reviewer #2: Yes

3. Have the authors made all data underlying the findings in their manuscript fully available?

Reviewer #1: Yes

Reviewer #2: Yes

4. Is the manuscript presented in an intelligible fashion and written in standard English?

Reviewer #1: Yes

Reviewer #2: Yes

5. Review Comments to the Author

Reviewer #1: GENERAL CONCEPT COMMENTS

The manuscript presents a valuable study investigating the accuracy of wrist-based heart rate monitors across different skin tones and exercise intensities. The research is well-structured and provides significant insights into how skin tone may affect heart rate measurement accuracy, particularly during varying exercise intensities. The discussion effectively contextualizes the findings within the existing literature and highlights the clinical relevance, particularly for individuals with darker skin tones and those at higher risk for cardiovascular conditions.

Overall, the study makes an important contribution to the field, and the findings have practical implications for improving the accuracy of heart rate monitors across diverse populations. Further refinement of some sections and attention to detail will enhance the manuscript's clarity and impact.

To enhance the depth of your manuscript, you might consider including references to additional studies that explore related aspects of heart rate monitoring and physiological assessments. These studies could provide valuable context and further support the findings of your research. Including such references might offer readers a more comprehensive understanding of the topic and its broader implications (Sampaio, T., Morais, J. E., & Bragada, J. A. (2024). StepTest4all: Improving the Prediction of Cardiovascular Capacity Assessment in Young Adults. Journal of Functional Morphology and Kinesiology, 9(1), 30. And Bragada, J. A., Magalhães, P. M., São-Pedro, E., Bartolomeu, R. F., & Morais, J. E. (2024). Net Heart Rate for Estimating Oxygen Consumption in Active Adults. Journal of Functional Morphology and Kinesiology, 9(2), 66.).

SPECIFIC COMMENTS

1. ABSTRACT

The abstract is well-structured, providing a brief overview of the background, methods, results, and conclusions.

2. INTRODUCTION

Please see this reference about devices’ metrological characteristics: Bouillod, A., Soto-Romero, G., Grappe, F., Bertucci, W., Brunet, E., & Cassirame, J. (2022). Caveats and recommendations to assess the validity and reliability of cycling power meters: A systematic scoping review. Sensors, 22(1), 386.

Specific textual recommendations for accuracy and precision:

(P.11 l.37 to 52 and 53 to 62) – The transition between the discussion on the general importance of HR monitoring during exercise and the specific focus on PPG technology across skin tones is somewhat abrupt. The manuscript would benefit from a smoother transition, perhaps by first addressing the general challenges of HR monitoring technologies before narrowing down to the specific issue of skin tone.

(P.11 l.53 to 62) – While the impact of skin tone on PPG accuracy is mentioned, there is insufficient depth in the review of existing literature. The paragraph should include references to key studies that have previously explored this issue, offering a more comprehensive context for the reader.

(P.12 l.60 to 62) – The issue of skin tone affecting PPG accuracy is mentioned, but the introduction does not sufficiently explain why this is a significant problem. Providing context, such as discussing the implications for diverse populations or the potential for health disparities, would make the rationale for the study clearer.

METHODS

Specific textual recommendations for accuracy and precision:

(P.13 l.86 to 101) – The subsection titled "Study Procedures" provides a general overview of the experimental protocol. However, the clarity and comprehensibility of this section would be greatly enhanced by dividing the content into more specific subsections.

(P.14 l.118) – It is suggested that the authors present examples of each skin tone category used in the study. This could include representative images or a clear description of how skin tones were classified.

(P.15 l.145) – The choice of a linear mixed-effects model is appropriate given the study's design. However, the manuscript does not discuss the assumptions underlying this model or whether these assumptions were tested (e.g., normality of residuals, homoscedasticity).

3. RESULTS

Specific textual recommendations for accuracy and precision:

(P.17 l.174) - The authors provide detailed information about the sample recruitment process and the characteristics of the participants in the Results section. While this information is crucial, it would be more appropriately placed in the Methods section.

(P.17 l.174) – Additionally, it is suggested that the authors include a graphical representation of the sample recruitment process, such as a flowchart or diagram.

(P.17 l.188) – The characteristics of the sample, currently presented in the Results section, should also be reallocated to the Methods section.

(P.17 l.188) – Regarding Table 1, the presentation of "mean and SD" (standard deviation) should be adjusted. Currently, it appears in the line of the variable; however, it would be clearer and more conventional to include "mean ± SD" in the column header where the data is presented.

(P.17 l.188) – Additionally, the meaning of "SD" should be clarified in a footnote to the table.

(P.22 l.241) – Provide Figure 1 with more quality.

4. DISCUSSION

The discussion section of the manuscript is well-written and effectively addresses the study's findings and their implications. The authors effectively contextualize their results within the existing literature, noting how their study advances understanding by controlling for exercise intensity—a factor that has not always been considered in previous research. The clinical relevance of the findings is well-articulated, with a strong emphasis on the practical implications for individuals with darker skin tones and those at risk of cardiovascular disease. Future research directions are appropriately suggested, focusing on the need for further investigation into correction algorithms and testing across diverse populations.

(P.23 l.261) – Add a period after the reference and check for similar instances throughout the manuscript.

5. REFERENCES

Verify that all references are formatted according to the journal guidelines.

Reviewer #2: The authors should provide a definition of heart rate reserve.

Referring to line 82, the fact that the authors spoke of a “primary aim” implies that there is at least a secondary one: what is it? Or do the authors mean something different? I encourage the authors to clearly express the aim/s of their study.

How were the HR measurements from the two devices synchronized to be considered time-matched?

At line 196 the measurement unit is missing. I encourage the authors to always express it.

An aspect should be clarified at line 126: the authors stated that they took the lowest HR observed after five minutes of quiet sitting as resting HR. If they took the minimum HR, they considered a time interval of observation after (then subsequently) 5 minutes of quite sitting: what is the time duration of this interval?

I suggest not to take for granted concepts and meaning of parameters (for example, alpha).

Explaining the secondary analysis, the authors stated that they “examined the accuracy in resting HR measurements across different skin tones”, and this seems in contradiction with lines 168-170.

Tables can be better described in their content and readers should be better guided in their interpretation.

In the Bland-Altman plots the indication of the x axis is missing.

In the discussion, the authors stated that “the small study sample size may have increased the possibility of Type 1 Errors”. Can the authors better explain why? What about Type 2 Errors?

6. PLOS authors have the option to publish the peer review history of their article (what does this mean?). If published, this will include your full peer review and any attached files.

Reviewer #1: No

Reviewer #2: No

---

## [Author Response · Author response to Decision Letter 0]

16 Oct 2024

Note, we have uploaded a formatted copy of the Response to Reviewers in the Attach File section already. We also have pasted the responses below.

October 11, 2024

Manuscript Title: Validity of heart rate measurements in wrist-based monitors across skin tones during exercise

Manuscript ID: PONE-D-24-32801

Dear Dr. Emily Chenette and Reviewers,

Thank you reviewing our manuscript and for the opportunity to respond to the comments for re-submission.

The following content includes a copy of changes requested as per journal requirements and of each reviewer’s comments, followed by our response under each comment. The page and line numbers of the revisions refer to the tracked changes version of the revised manuscript. 

Sincerely,

Janice Eng, PT, PhD

Professor

Centre for Aging SMART, Vancouver Coastal Health

4255 Laurel Street, Vancouver BC, V5Z 2G9 Canada

Department of Physical Therapy, Faculty of Medicine

University of British Columbia

Response: We have made specific edits to align with PLOS ONE’s style requirements. This includes all heading levels, and figure and table titles and captions throughout the manuscript. 

Response: The statistical analysis code for the data analysis is available on the corresponding author’s Borealis (A Canadian Dataverse Repository) account (https://doi.org/10.5683/SP3/LLD99A). An updated Data Availability Statement has been included to the submission. 

“Funding for the current study was supported by the Canada Research Chairs Program (to J Eng), Michael Smith Health BC Postdoctoral Award (to S Hung), and the Canadian Institutes of Health Research Foundation Grant (FDN 143340; to J Eng).”

Response: The role of the funder has been included in the updated cover letter. Thank you for amending the Role of Funder statement on the online submission form.

“Funding for the current study was supported by the Canada Research Chairs Program (to J Eng), Michael Smith Health BC Postdoctoral Award (to S Hung), and the Canadian Institutes of Health Research Foundation Grant (FDN 143340; to J Eng). The study team would like to thank the study participants and Asahi Ng, Jackson Liu, and Nadine Akbarali for contributing to participant recruitment and data collection.”

“Funding for the current study was supported by the Canada Research Chairs Program (to J Eng), Michael Smith Health BC Postdoctoral Award (to S Hung), and the Canadian Institutes of Health Research Foundation Grant (FDN 143340; to J Eng).”

Response: We have removed the funding statement from the manuscripts text acknowledgements. No changes to the Funding Statement are required. 

Reviewer's Responses to Questions

Reviewer #1:

To enhance the depth of your manuscript, you might consider including references to additional studies that explore related aspects of heart rate monitoring and physiological assessments. These studies could provide valuable context and further support the findings of your research. Including such references might offer readers a more comprehensive understanding of the topic and its broader implications (Sampaio, T., Morais, J. E., & Bragada, J. A. (2024). StepTest4all: Improving the Prediction of Cardiovascular Capacity Assessment in Young Adults. Journal of Functional Morphology and Kinesiology, 9(1), 30. And Bragada, J. A., Magalhães, P. M., São-Pedro, E., Bartolomeu, R. F., & Morais, J. E. (2024). Net Heart Rate for Estimating Oxygen Consumption in Active Adults. Journal of Functional Morphology and Kinesiology, 9(2), 66.).

Response: Thank you to the reviewer for suggesting these references and associated topics. We included the reviewer’s second suggested reference (Bragada, J. A., Magalhães, P. M., São-Pedro, E., Bartolomeu, R. F., & Morais, J. E. (2024). Net Heart Rate for Estimating Oxygen Consumption in Active Adults. Journal of Functional Morphology and Kinesiology, 9(2), 66.) as it specifically pertains the heart rate measurements in the estimation of oxygen consumption, and highlights the importance of accurate heart rate in monitoring exercise response, exercising testing and estimating oxygen consumption. 

Page 3; Line 48: The popularity of wearable fitness trackers to monitor HR has increased in recent years for consumers in the general public, healthcare, and research settings (5), including monitoring HR for exercise prescription or estimating oxygen consumption during exercise testing (6).

2. INTRODUCTION

Please see this reference about devices’ metrological characteristics: Bouillod, A., Soto-Romero, G., Grappe, F., Bertucci, W., Brunet, E., & Cassirame, J. (2022). Caveats and recommendations to assess the validity and reliability of cycling power meters: A systematic scoping review. Sensors, 22(1), 386.

Response: Thank you for this suggestion, and for outlining the different properties of validity to note. In the first instance of describing validity, we have clarified that we are referring to accuracy of heart rate measurements. 

Page 3; Line 53: For these reasons, wireless, ECG-based chest strap HR monitors have grown in popularity and are valid (i.e., accuracy of measuring heart rate) alternatives to ECG (6).

(P.11 l.37 to 52 and 53 to 62) – The transition between the discussion on the general importance of HR monitoring during exercise and the specific focus on PPG technology across skin tones is somewhat abrupt. The manuscript would benefit from a smoother transition, perhaps by first addressing the general challenges of HR monitoring technologies before narrowing down to the specific issue of skin tone.

Response: We added general context of challenges with PPG technology for heart rate measurement prior to discussing skin tone.

Page 3; line 66: Accurate measurements of HR using PPG devices is highly dependent on PPG signal accuracy, which can be influenced by multiple factors, including type of sensors used (example, shape of sensors and light intensity), post-processing algorithms used, the deformation of the skin when device is worn, vascular aging (example, arterial stiffness), and excessive movement (15). 

(P.11 l.53 to 62) – While the impact of skin tone on PPG accuracy is mentioned, there is insufficient depth in the review of existing literature. The paragraph should include references to key studies that have previously explored this issue, offering a more comprehensive context for the reader.

Response: In our initial submission, we included a key reference to the systematic review completed by Koerber and colleagues, who used rigorous systematic review methods to investigate the impact of skin tone on accuracy of heart rate measures using wrist-based devices. To provide more comprehensive context to the reader, we provided more details on interpretation of the existing literature to date by Koerber and colleagues, such as elaborating on the potential reasons for inconsistent results within the existing literature (inadequate sample size, non-standardized use of skin tone categorization) and how our study adds to the literature. 

Page 4; line 76: Another systematic review (10 studies, 469 participants, 26 different devices) rigorously examined the available literature on the impact of skin tone on the accuracy of HR measurements using wrist-based PPG devices (18). The authors concluded that there are conflicting results in HR measurement accuracy across different skin tones, with four studies observing lower accuracy with darker skin tones, four observing no difference, and two having mixed results (18). The authors attributed the mixed results partly due to inadequate sample sizes, use of multiple devices for comparison, and inconsistent methods with categorizing skin tones (example, non-standardized self-reported race and ethnicity) (18). 

(P.12 l.60 to 62) – The issue of skin tone affecting PPG accuracy is mentioned, but the introduction does not sufficiently explain why this is a significant problem. Providing context, such as discussing the implications for diverse populations or the potential for health disparities, would make the rationale for the study clearer.

Response: We provided more context by introducing the issue with HR inaccuracies when monitoring exercise for clinical population

Page 5; line 94: This potential source of inaccuracy could have important implications for use across ethnically-diverse populations, especially when HR inaccuracies could pose a safety risk, particularly in clinical settings for people with darker skin (18,24). In clinical settings, such as people with cardiovascular diseases, monitoring HR during exercise is recommended to ensure people are safely exercising at HR below the onset of ischemic symptoms and at an adequate intensity to maximize exercise adaptations and benefits (31). People with darker skin tones (example, Black and Hispanic ethnic backgrounds) are at higher risk of cardiovascular diseases (32), and inaccurate HR measurements using PPG-based devices would result in potential health disparities with exercise safety and treatment efficacy in this clinical population.

METHODS

(P.13 l.86 to 101) – The subsection titled "Study Procedures" provides a general overview of the experimental protocol. However, the clarity and comprehensibility of this section would be greatly enhanced by dividing the content into more specific subsections.

Response: 

Page 7; 143 and Page 8; line 158::“Skin tone categorization” and “Resting and exercise testing heart rate” subsections have been added.

(P.14 l.118) – It is suggested that the authors present examples of each skin tone category used in the study. This could include representative images or a clear description of how skin tones were classified.

Response: We included a description of the skin tone categories in table form (Table 1; all tables number adjusted thereafter). 

Page 7; line 146: …the Fitzpatrick Scale, which categorizes skin tones across 6 types based on colour and ultraviolet radiation response (36). Table 1 describes the 6 Fitzpatrick Scale skin tone categories (37). 

Table 1 has been added to Page 7; line 156. 

(P.15 l.145) – The choice of a linear mixed-effects model is appropriate given the study's design. However, the manuscript does not discuss the assumptions underlying this model or whether these assumptions were tested (e.g., normality of residuals, homoscedasticity).

Response: We thank the reviewer for the opportunity to expand on the appropriateness of linear mixed effects models in the current study. We tested the assumptions of our model (results found in Supporting Information File 1, S1_File.pdf), and observed mild heteroskedasticity of variance and mild positive skewness in our fixed effects residuals. Linear mixed effects models have been shown to be robust to such violations of distributional assumptions. We included three key references in support of the robustness of the linear mixed effects models: 

45. Harrison XA, Donaldson L, Correa-Cano ME, Evans J, Fisher DN, Goodwin CE, et al. A brief introduction to mixed effects modelling and multi-model inference in ecology. PeerJ. 2018 May 23;6:e4794. (

46. Schielzeth H, Dingemanse NJ, Nakagawa S, Westneat DF, Allegue H, Teplitsky C, et al. Robustness of linear mixed-effects models to violations of distributional assumptions. Methods Ecol Evol. 2020;11(9):1141–52. 

47. Jacqmin-Gadda H, Sibillot S, Proust C, Molina JM, Thiébaut R. Robustness of the linear mixed model to misspecified error distribution. Comput Stat Data Anal. 2007 Jun 15;51(10):5142–54. 

We updated the manuscript in the Statistical Analysis to include the results of assumption testing for our linear mixed effects model:

Page 10; line 203: We tested the linear mixed effects model assumptions (see S1 File for assumptions results). We observed mild heteroskedasticity of variance (S1 Fig in S1 File) and mild positive skewness in our fixed effects residual distribution (S2 Fig and S3 Fig in S1 File). Linear mixed effects models are robust to violation to distributional assumptions and heteroscedasticity and remains an appropriate model for our study data (45–47).

Page 28; line 538: We added the Supporting Information List at the end of the manuscript.

3. RESULTS

(P.17 l.174) - The authors provide detailed information about the sample recruitment process and the characteristics of the participants in the Results section. While this information is crucial, it would be more appropriately placed in the Methods section.

Response: We thank the reviewer for the suggestion. We have outlined procedures we followed to recruit study participants in the Methods section. We believe that the outcome of the participant recruitment procedures we followed, including the sample recruitment process (number of people who expressed interested, were screened and recruited, and later excluded), and characteristics of the participants, is better suited in the Results section. 

(P.17 l.174) – Additionally, it is suggested that the authors include a graphical representation of the sample recruitment process, such as a flowchart or diagram.

Response: A recruitment flow chart has been included, and is now labelled as Fig1 (all figure numbering has been updated).

Figure caption has been added to Page 11; line 236: Fig1. Study recruitment process flowchart.

(P.17 l.188) – The characteristics of the sample, currently presented in the Results section, should also be reallocated to the Methods section.

Response: We thank the reviewer for the suggestion. We believe that the outcome of the participant recruitment procedures, including the characteristics of the participants, is better suited in the Results section.

(P.17 l.188) – Regarding Table 1, the presentation of "mean and SD" (standard deviation) should be adjusted. Currently, it appears in the line of the variable; however, it would be clearer and more conventional to include "mean ± SD" in the column header where the data is presented.

Response: In all tables, we have changed all presentations of “mean (SD)” to “mean ± SD” to align with convention, where applicable. We did not include this in the column header for this table (now Table 2), given that the “Female” variable is not presented in means and standard deviations. 

(P.17 l.188) – Additionally, the meaning of "SD" should be clarified in a footnote to the table.

Response: This has been clarified in the footnotes of all tables

---

## [Decision Letter · Decision Letter 1]

11 Nov 2024

PONE-D-24-32801R1Validity of heart rate measurements in wrist-based monitors across skin tones during exercisePLOS ONE

Dear Dr. Eng,

Thank you for submitting your manuscript to PLOS ONE. After careful consideration, we feel that it has merit but does not fully meet PLOS ONE’s publication criteria as it currently stands. Therefore, we invite you to submit a revised version of the manuscript that addresses the points raised during the review process.

We look forward to receiving your revised manuscript.

Kind regards,

Agnese Sbrollini

Academic Editor

PLOS ONE

Journal Requirements:

Reviewers' comments:

Reviewer's Responses to Questions

**Comments to the Author**

1. If the authors have adequately addressed your comments raised in a previous round of review and you feel that this manuscript is now acceptable for publication, you may indicate that here to bypass the “Comments to the Author” section, enter your conflict of interest statement in the “Confidential to Editor” section, and submit your "Accept" recommendation.

Reviewer #1: All comments have been addressed

Reviewer #2: (No Response)

2. Is the manuscript technically sound, and do the data support the conclusions?

Reviewer #1: Yes

Reviewer #2: Yes

3. Has the statistical analysis been performed appropriately and rigorously? 

Reviewer #1: Yes

Reviewer #2: Yes

4. Have the authors made all data underlying the findings in their manuscript fully available?

Reviewer #1: Yes

Reviewer #2: Yes

5. Is the manuscript presented in an intelligible fashion and written in standard English?

Reviewer #1: Yes

Reviewer #2: Yes

6. Review Comments to the Author

Reviewer #1: I congratulate the authors for improving the manuscript as advised. In my opinion, this can be published in the present form.

Reviewer #2: Regarding synchronization, probably an artifact performed simultaneously (e.g., the simultaneous execution of a pressure) on the two devices could ensure more accurate timing. However, if the investigator in charge of timing ensures accuracy and attention, good synchronization can be achieved.

Can the authors better explain the difference between “alpha” and “power”?

The authors re-referenced the results sentences back to the respective tables to improve interpretation, but I suggest checking table numbers.

The current Table 6 should be further described. What is the role of the intercept? What can we deduce from its value? And what about random effects? “ICC” is not even defined (as well as the other random effects). My suggestion is not to take concepts for granted by leaving too much to the reader's interpretation, even for “standard” concepts or parameters. Every content in the table should at least be defined/described. What is shown in the table? Even the acronym “HR” is fairly standard, especially for certain areas of research, but the authors have rightly defined it in Abstract (and should also define it when first used in the text)

In the current figure 2, I suggest specifying that on x axis means are reported and on y axis differences are reported.

7. PLOS authors have the option to publish the peer review history of their article (what does this mean?). If published, this will include your full peer review and any attached files.

Reviewer #1: No

Reviewer #2: No

---

## [Author Response · Author response to Decision Letter 1]

12 Dec 2024

November 14, 2024

Manuscript Title: Validity of heart rate measurements in wrist-based monitors across skin tones during exercise

Manuscript ID: PONE-D-24-32801

Dear Dr. Emily Chenette and Reviewers,

Thank you reviewing our manuscript and for the opportunity to respond to the comments for re-submission.

The following content includes a copy of changes requested as per journal requirements and of each reviewer’s comments, followed by our response under each comment. The page and line numbers of the revisions refer to the tracked changes version of the revised manuscript. 

Sincerely,

Janice Eng, PT, PhD

Professor

Centre for Aging SMART, Vancouver Coastal Health

4255 Laurel Street, Vancouver BC, V5Z 2G9 Canada

Department of Physical Therapy, Faculty of Medicine

University of British Columbia

Journal Requirements:

Reviewer #1: I congratulate the authors for improving the manuscript as advised. In my opinion, this can be published in the present form.

Response: We thank reviewer 1 for their helpful feedback to improve the manuscript. 

Reviewer #2: 

Can the authors better explain the difference between “alpha” and “power”?

Response: We further explained the concepts of alpha and power in the statistical analysis to help distinguish the two concepts:

Page 9; line 180: 0.05 alpha (level of significance; 5% probability of incorrectly rejecting the null hypothesis or falsely detecting an effect when there is no true effect [Type 1 Error]).

Page 10; Line 199: 80% power (probability of correctly rejecting the null hypothesis; 80% chance of detecting statistical significance when a true effect is present)

The authors re-referenced the results sentences back to the respective tables to improve interpretation, but I suggest checking table numbers.

Response: We corrected references to Table 3 and Table 4 in the results (Page 12; line 241 and 242).

The current Table 6 should be further described. What is the role of the intercept? What can we deduce from its value? And what about random effects? “ICC” is not even defined (as well as the other random effects). My suggestion is not to take concepts for granted by leaving too much to the reader's interpretation, even for “standard” concepts or parameters. Every content in the table should at least be defined/described. What is shown in the table? Even the acronym “HR” is fairly standard, especially for certain areas of research, but the authors have rightly defined it in Abstract (and should also define it when first used in the text)

Response: Thank you for this suggestion to clarify the contents and concepts of Table 6. We have clarified interpretation of the intercept, random effects, and intraclass correlation coefficient (ICC). We included these details in the table footer. 

Page 17; Lines 284 to 289: Note: the intercept indicates the absolute error in heart rate without including effects of other skin tones (i.e., the effect of light skin tone only). For random effects: an σ2 value of 114.19 bpm indicates a 10.69 bpm (square root of 114.19) of absolute error explained by within-participant variance; a τ00 participants value of 26.10 bpm indicates 5.11 bpm (square root of 26.10) between-participant variance; a 0.19 intraclass correlation coefficient indicates that 19% of the total variance of absolute error of heart rate is attributed to between-participant variance. 

We have also clarified that our absolute errors are for heart rate measures in the title and/or appliable column heading for Table 6 (Page 16), along with Table 3 (page 13), Table 4 (Page 13), and Table 5 (Page 14). 

HRR has already been defined as heart rate reserve in the table footer.

In the current figure 2, I suggest specifying that on x axis means are reported and on y axis differences are reported.

Response: We amended the axis x axis labels to “Mean of Polar H10 and Fitbit Charge 5” and y axis labels to “Difference between Polar H10 - Fitbit Charge 5”

---

## [Decision Letter · Decision Letter 2]

22 Jan 2025

Validity of heart rate measurements in wrist-based monitors across skin tones during exercise

PONE-D-24-32801R2

Dear Dr. Eng,

We’re pleased to inform you that your manuscript has been judged scientifically suitable for publication and will be formally accepted for publication once it meets all outstanding technical requirements.

Kind regards,

Agnese Sbrollini

Academic Editor

PLOS ONE

Additional Editor Comments (optional):

Reviewers' comments:

Reviewer's Responses to Questions

**Comments to the Author**

1. If the authors have adequately addressed your comments raised in a previous round of review and you feel that this manuscript is now acceptable for publication, you may indicate that here to bypass the “Comments to the Author” section, enter your conflict of interest statement in the “Confidential to Editor” section, and submit your "Accept" recommendation.

Reviewer #1: All comments have been addressed

Reviewer #2: All comments have been addressed

2. Is the manuscript technically sound, and do the data support the conclusions?

Reviewer #1: Yes

Reviewer #2: Yes

3. Has the statistical analysis been performed appropriately and rigorously? 

Reviewer #1: Yes

Reviewer #2: Yes

4. Have the authors made all data underlying the findings in their manuscript fully available?

Reviewer #1: Yes

Reviewer #2: Yes

5. Is the manuscript presented in an intelligible fashion and written in standard English?

Reviewer #1: Yes

Reviewer #2: Yes

6. Review Comments to the Author

Reviewer #1: nothing to declare. XXXXXXXXXXXXXXXXXXXXXXXXXXXXXXXXXXXXXXXXXXXXXXXXXXXXXXXXXXXXXXXXXXXXXXXXXXXXXXXX

Reviewer #2: The authors addressed all the reviewer's observations and improved the manuscript as advised. My opinion is that the manuscript can be published in the present form.

7. PLOS authors have the option to publish the peer review history of their article (what does this mean?). If published, this will include your full peer review and any attached files.

Reviewer #1: No

Reviewer #2: No

---

## [Editor Report · Acceptance letter]

30 Jan 2025

PONE-D-24-32801R2 

PLOS ONE

Dear Dr. Eng, 

I'm pleased to inform you that your manuscript has been deemed suitable for publication in PLOS ONE. Congratulations! Your manuscript is now being handed over to our production team.

Kind regards, 

on behalf of

Dr. Agnese Sbrollini 

Academic Editor

PLOS ONE